# Unlocking Exocentric Video-Language Data for Egocentric Video Representation Learning

## Abstract

We present EMBED (Egocentric Models Built with Exocentric Data), a framework designed to mine video-language data from exocentric sources for egocentric video representation learning. Large-scale exocentric data covers diverse activities with significant potential for egocentric learning, but inherent disparities between egocentric and exocentric data pose challenges in utilizing one view for the other seamlessly. In this study, we propose leveraging hand-object interactions and language narratives as cues to incorporate exocentric data into egocentric training. Specifically, we focus on identifying specific video clips that emphasize hand-object interactions and pairing them with action-focused language narrations. By applying our framework to exocentric datasets such as HowTo100M, we construct datasets thar are effective for egocentric video-language pretraining. Our extensive evaluations reveal that EMBED achieves state-of-the-art performance across various egocentric downstream tasks, including a 4.7% absolute improvement in multi-instance retrieval on the Epic-Kitchens-100 benchmark and a 6.2% improvement in classification on the EGTEA benchmark in zero-shot settings. Furthermore, EMBED enables egocentric video-language models to perform competitively in exocentric tasks. Finally, we showcase EMBED's application across various exocentric datasets, exhibiting strong generalization capabilities when applied to different exocentric datasets.

## 1 Introduction

Egocentric video understanding has become a crucial research field, notably impacting areas like augmented reality, personal assistants, and robotics. The curation of egocentric video-language datasets (Damen et al., 2018; Grauman et al., 2022) has catalyzed progress in this domain, enabling significant advancements in video understanding through the use of video-language pretraining (Lin et al., 2022; Zhao et al., 2023; Pramanick et al., 2023; Ashutosh et al., 2023).

While there are several egocentric video-language datasets available, exocentric datasets encompass a broader range of human activities, which can be potentially used to enhance egocentric representation learning. Nonetheless, there is a noticeable domain gap that challenges seamless utilization (Li et al., 2021b; Lin et al., 2022; Wang et al., 2023). This gap manifests in two dimensions: (1) **video content**, where egocentric videos predominantly capture close-up hand-object interactions, offering a detailed perspective from the camera wearer's point of view, while exocentric videos capture a broader scene including both the subjects' actions and their contextual environment; (2) **the language narration style** differs significantly, with egocentric videos often accompanied by action-focused, human-annotated narrations and exocentric videos relying on less accurate automatic transcriptions. Consequently, few have found effective ways to best utilize videos of one viewpoint for another, often resorting to simply finetuning models trained on separate viewpoints (Zhao et al., 2023) or training models with egocentric data only (Lin et al., 2022; Pramanick et al., 2023; Wang et al., 2023). Notably, Ego-Exo (Li et al., 2021b) proposes to leverage exocentric video classification data for egocentric learning by distilling egocentric cues from exocentric data into video encoders. However, this method is focused on video classification data with categorical labels, making it challenging to adapt for video-language pretraining with flexible language narrations.

In this work, we present our method for automatically mining exocentric video-language data for egocentric video representation learning. As illustrated in Figure 1, despite their distinct viewpoints,

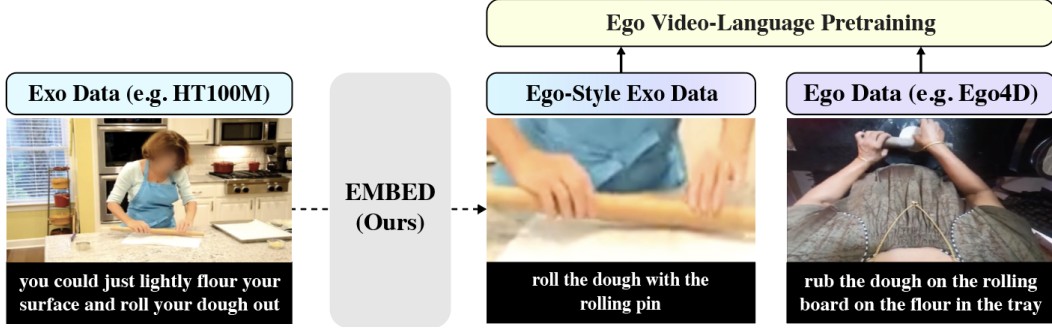

**Figure 1:** Despite the domain difference, exocentric data can contain egocentric cues such as hand-object interaction information in vision and language modalities. Our EMBED method leverages these cues, constructing video-language data for egocentric representation learning from exocentric sources.

exocentric and egocentric data can share similar hand-object interaction (HOI) information reflected in both the vision and language modalities, which can be potentially leveraged to improve egocentric learning. Motivated by this, we propose EMBED (**E**gocentric **M**odels **B**uilt with **E**xocentric **D**ata), a method designed to construct egocentric-style video-language data from exocentric sources by using egocentric cues. First, we identify and utilize HOI information to curate egocentric-relevant video clips from the exocentric dataset. This process involves selecting video clips that prominently feature active HOIs and cropping out the HOI regions spatially. This targeted approach allows for a more precise extraction of egocentrically relevant information from exocentric sources. Second, we perform narration generation to pair each video with narrations styled after egocentric data. We implement this through two models: 1) **ego narrator**, a narration generation model trained on egocentric data. This model is utilized to generate narrations for exocentric videos, ensuring the output mirrors the egocentric style; 2) **exo-to-ego rephraser**, which employs a large language model for in-context learning. This model translates existing exocentric narrations into the egocentric style, effectively adapting the language to match the egocentric context. By combining the video curation and narration generation strategies, we construct new video-language data from exocentric sources that is tailored for egocentric representation learning.

We perform extensive evaluations of EMBED across multiple egocentric video downstream tasks. Specifically, we first demonstrate that integrating existing exocentric data (*e.g.*, HowTo100M (Miech et al., 2019b; Han et al., 2022)) into egocentric pretraining is suboptimal and can sometimes even hurt the model performance. In contrast, applying our proposed method and then combining egocentric and exocentric data can substantially improve the model performance, setting the state of the art on a wide range of challenging downstream tasks. Notably, EMBED achieves an absolute improvement of 4.7% on the Epic-Kitchens-100 multi-instance retrieval and 6.2% on the EGTEA classification benchmarks. In addition, training with both egocentric and exocentric data yields benefits beyond egocentric tasks, enabling our model to achieve comparable performance than models trained exclusively with exocentric data in tasks such as UCF-101 (Soomro et al., 2012) and HMDB-51 (Kuehne et al., 2011). Moreover, experiments suggest that EMBED exhibits strong generalization when transferring from different exocentric datasets, including HowTo100M (Miech et al., 2019b), Kinetics-700 (Carreira et al., 2019), Something-Something v2 (Goyal et al., 2017), and COIN (Tang et al., 2019).

In summary, our contributions are: (1) we introduce a framework that connects exocentric and egocentric data with hand-object interaction and language narration information; (2) we propose data mining strategies that function in both vision (video temporal selection and spatial zoom-in) and language (rephrasing and generation) modalities within this framework, resulting in new video-language data sourced from exocentric data tailored for egocentric learning; (3) we demonstrate the effectiveness of our framework across benchmarks.

## 2 METHOD: EMBED

**Formulation.** Formally, a video is split into a set of non-overlapping short clips. Each video clip $x$ consists of several frames $\langle f_{x_1}, \cdots, f_{x_k} \rangle$, and is often paired with a free-form language

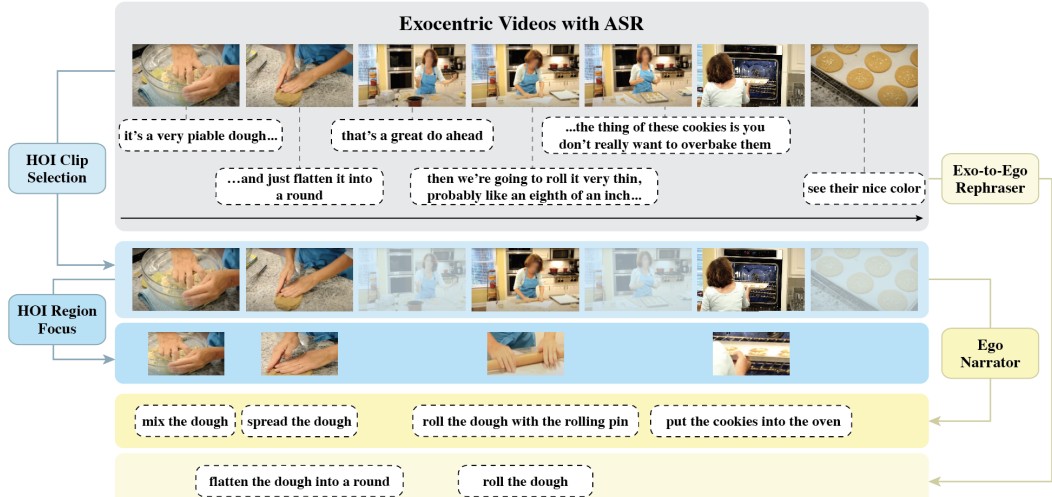

**Figure 2:** Given an exocentric dataset, EMBED selects video clips featuring hand-object interactions (HOI) and further refines these selections by focusing on HOI regions to offer a close-up view. Additionally, we pair each exocentric clip with narrations emphasizing human actions, akin to those in egocentric data. This is achieved by using a narrator model trained on egocentric data; also, we employ an exo-to-ego rephraser model that converts existing sentences into action-oriented narrations that reflect an egocentric perspective.

annotation $y$. The language annotation is either automatically transcribed by a model given audio (e.g. HowTo100M (Miech et al., 2019b)) or manually annotated describing the human actions (e.g. Ego4D (Grauman et al., 2022)). Given both egocentric and exocentric datasets of (video clip, language narration) pairs, denoted as $(\mathcal{X}^{ego}, \mathcal{Y}^{ego})$ and $(\mathcal{X}^{exo}, \mathcal{Y}^{exo})$, our goal is to transform $(\mathcal{X}^{exo}, \mathcal{Y}^{exo})$ so that its style is similar to that of $(\mathcal{X}^{ego}, \mathcal{Y}^{ego})$.

**Overview.** While egocentric and exocentric video-language datasets share similar data formats, they differ in terms of video content and language narration style, preventing effective training on their concatenated data. To solve the issue, as shown in Figure 2, our EMBED method first curates egocentric-relevant videos $\mathcal{X}^{exo-ego}$ from exocentric data (Section 2.1), then pairs each video with its corresponding egocentric-style language narration $\mathcal{Y}^{exo-ego}$ (Section 2.2). Afterwards, we train a video-language model on the egocentric and transformed exocentric data $(\mathcal{X}^{ego}, \mathcal{Y}^{ego}) \oplus (\mathcal{X}^{exo-ego}, \mathcal{Y}^{exo-ego})$, and the learned representations can be used for downstream applications.

## 2.1 VIDEO CLIP CURATION

In this section, we provide a detailed explanation of our approach to curating video clips that are highly relevant to egocentric scenarios from an exocentric dataset. This curation process effectively collects egocentric-style videos from exocentric sources.

**Temporal Selection of HOI Video Clips.** One of the key challenges in aligning exocentric data with the egocentric context is the inherent diversity of content within exocentric videos. Exocentric videos may contain various actions, including irrelevant ones such as individuals looking around or engaging in unrelated activities (Miech et al., 2019a;b; Han et al., 2022). This diversity complicates the alignment of exocentric data with the format of egocentric data, which primarily emphasizes hand-object interactions (HOI).

To tackle this challenge, we introduce a strategy for selecting video clips that emphasize the HOI content. We start by uniformly sampling video clips from the entire exocentric dataset, each spanning 5 seconds. Subsequently, we employ a robust off-the-shelf hand-object detector (Shan et al., 2020) to densely extract regions of HOI from all the video clips. Specifically, for each video clip, we sample 4 frames and use the hand-object detector to extract bounding boxes for the hand, object, and HOI regions along with their prediction probabilities within those frames.

Once the HOI regions are extracted, we assess the relevance of each video clip, denoted as $x = \langle f_{x_1}, \cdots, f_{x_k} \rangle$, using the following scoring function:

$$HOI\_score(x) = \frac{1}{k} \sum_i (HOI(f_{x_i}) + AVG\_HP(f_{x_i})), \tag{1}$$

where $HOI(\cdot)$ is a binary function indicating the presence of hand-object interaction in a video frame, and $AVG\_HP(\cdot)$ represents the average probability of all the detected hands in that frame, which can indirectly capture how well a video clip depicts the hand-object interactions.

Subsequently, we rank the video clips based on their scores and select those with the highest scores to be included in our training dataset. These selected video clips from the exocentric dataset prominently feature hand-object interactions. Each chosen video clip is then paired with the corresponding language narration from the original dataset, provided that the narration's timestamp falls within the clip's time interval, and we illustrate how we transform the language narrations in the next section.

**Spatial Focus on HOI Regions.** In addition to the temporal selection, we propose a technique to further encourage the model's focus on hand-object interaction regions spatially. To achieve this, we extract and zoom in on the HOI regions within the temporally selected video clips. This approach, as in Figure 2, aligns the format of the curated videos more closely with that of egocentric data.

Based on the hand and object regions obtained during the temporal selection step, we perform cropping and zooming to isolate these specific regions, creating video clips that closely resemble the close-up hand-object interactions characteristic of egocentric data. First, we combine all the extracted hand and object bounding boxes from each frame to form their convex hull, resulting in a combined bounding box that covers all the hands and objects detected. During training, we randomly alternate between using the original video clip and its cropped, zoomed-in version, with an equal probability assigned to each selection.

This spatial selection strategy offers multiple advantages. It promotes similarity between the formats of egocentric and exocentric data, facilitating seamless integration. Furthermore, it implicitly encourages our models to focus on the hand-object interaction region, as video-language pretraining losses such as the contrastive learning loss align the representations of both the original video clip and the zoomed-in clip with the same language target. Additionally, it serves as a data augmentation strategy.

**Summary.** By combining temporal and spatial selection techniques on video clips from $\mathcal{X}^{exo}$, we curate a set of video clips $\mathcal{X}^{exo-ego}$ that are rich in HOIs and highly relevant to egocentric learning.

## 2.2 LANGUAGE NARRATION GENERATION

In this part, we present our method for pairing each curated exocentric video with egocentric-style narrations using both egocentric-style narration paraphrasing and generation. Different from exocentric narrations which are usually obtained from noisy automatically transcribed sentences, narrations in egocentric datasets are typically manually annotated and focused on human actions. We demonstrate our method through examples of pairing the videos in the HowTo100M dataset with narrations of the Ego4D style, and we will show the applicability to generalize the idea to other datasets in the experiment section.

**Exo-to-Ego Rephraser.** Exocentric narrations often comprise ASR (Automatic Speech Recognition) sentences that may include content irrelevant to egocentric representation learning. For example, sentences such as that *"i cannot wait to dig in and enjoy it on the outside"* lack visual alignment, making them less useful. Similarly, sentences like *"we're going to keep mixing it because you don't want your chocolate to stick to the bottom of your pot"* delve into the reasoning behind actions, which can divert the model's focus. In contrast, narrations in the Ego4D dataset are typically succinct, like "C turns on a light," primarily emphasizing human actions.

Our goal is to transform exocentric narrations into the egocentric style when applicable. For example, the sentence "i'm just gonna start by cutting it in half" will be transformed to "a person cuts it in half". To initiate the transformation of exocentric narrations, we can use large language models, and here we employ the Llama-2 model (Touvron et al., 2023). In order to adapt Llama-2 for our specific task, we begin by manually annotating a set of 10 examples comprising exocentric narrations and their

corresponding egocentric-style counterparts. These annotated pairs serve as our few-shot learning examples for in-context learning for Llama-2.[1] Llama-2, prompted with this annotated dataset, is then utilized as a paraphrasing tool to generate egocentric-style paraphrases of the exocentric narrations. We refer readers to Appendix for more details.

In practice, we observe that many sentences are not visually alignable. For instance, narrations may include background information that detracts from visual alignment, as noted by Han et al. (2022). Additionally, these non-visually alignable sentences often lack action information, making transformation using our rephraser impossible. To address this issue, we propose a method to filter out such sentences. Specifically, we fine-tune a text classification model, DeBERTa-v3 (He et al., 2021), using the HTM-Align dataset (Han et al., 2022). This dataset consists of a manually annotated collection of 80 videos and 5,021 sentences from HowTo100M, with each sentence tagged for its visual alignment with the corresponding video content. We utilize the fine-tuned DeBERTa-v3 model to filter out sentences that lack visual alignment. Subsequently, we process the visually alignable sentences through the Llama-2 model for style transformation.

This exo-to-ego narration transformation process effectively translates the original exocentric narrations into a more egocentric perspective, ensuring that the core information is retained while the style and viewpoint are adjusted to align with an egocentric narrative.

**Ego Narrator.** In addition to transforming existing exocentric narrations, we develop an egocentric-style narration generator as a separate component in our generation process. This generator is trained on a dataset containing purely egocentric data. Unlike the exo-to-ego narration paraphraser, which focuses on paraphrasing existing exocentric narrations, the generator's purpose is to create new egocentric-style narrations from scratch. Given an exocentric video clip, the narrator generator is capable of producing an egocentric-style narration based on the video content, ensuring that they are contextually relevant and consistent with an egocentric style. In this paper, we adopt the narrator model in LaViLa trained on Ego4D and use it to generate narrations on exocentric data.

Because the generated captions can sometimes be of low quality, we filter the low-quality samples based on the model's confidence scores, which are measured by perplexities, and filter any generations whose perplexity scores are lower than a threshold. In addition, because we find that the generation quality is more important than diversity as we will show in the experiment section, we propose to perform inference using beam sampling instead of nucleus sampling.

**Summary.** Our approach involves two independent components that obtain egocentric-style narrations from two different sources: the exo-to-ego rephraser for paraphrasing existing exocentric language narrations into an egocentric style, and the ego narrator for generating egocentric-style narrations directly from exocentric video content. These components pair exocentric videos with egocentric-style narrations, resulting in our curated egocentric-relevant data $(\mathcal{X}^{exo-ego}, \mathcal{Y}^{exo-ego})$.

### 2.3 TRAINING WITH OUR CURATED DATA

**Curated Dataset.** Using our method, we can construct new egocentric-style data sourced from exocentric data. For example, when we apply our method to the HTM-AA dataset (Han et al., 2022), which is a subset of HT100M containing around 247K videos and 3.3M video-narration pairs, we obtain a dataset consisting of 202K videos and 2.4M video-narration pairs in total, with each video clip containing HOI information detectable by (Shan et al., 2020). Out of the 2.4M video-narration pairs, approximately 1.7 million come from the generator, while around 700K are sourced from the rephraser.

**Joint Training.** We concatenate the original egocentric dataset $(\mathcal{X}^{ego}, \mathcal{Y}^{ego})$ with the curated exocentric data $(\mathcal{X}^{exo-ego}, \mathcal{Y}^{exo-ego})$. At each training step, we sample a batch of data from the concatenated dataset $\mathcal{B} \sim (\mathcal{X}^{ego}, \mathcal{Y}^{ego}) \oplus (\mathcal{X}^{exo-ego}, \mathcal{Y}^{exo-ego})$. In this paper, we train a video-language dual encoder model (Zhao et al., 2023) with the contrastive loss on the sampled batch $\mathcal{B}$ with InfoNCE:

$$\mathcal{L} = \frac{1}{|\mathcal{B}|} \sum_{(x_i, y_i) \in \mathcal{B}} [\log \frac{e^{s(x_i, y_i)/\tau}}{\sum_{y_j \in B} e^{s(x_i, y_j)/\tau}} + \log \frac{e^{s(x_i, y_i)/\tau}}{\sum_{x_k \in B} e^{s(x_k, y_i)/\tau}}], \tag{2}$$

---

[1]We provide the specific prompt in Appendix.

| Model | Pretrain Data | EK-100 MIR | | EK-100 CLS | | EGTEA | | EgoMCQ | |
|---|---|---|---|---|---|---|---|---|---|
| | | mAP | nDCG | top-1 acc. | top-5 acc. | mean acc. | top acc. | intra acc. | inter acc. |
| EgoVLP | Ego4D | 16.6 | 23.1 | - | - | - | - | 57.2 | 90.6 |
| Xu et al. (2024) | Ego4D | 31.6 | 34.9 | - | - | - | - | 54.2 | 92.7 |
| EgoVLPv2-B | Ego4D | 26.7 | 29.1 | - | - | - | - | 60.9 | 91.0 |
| LaViLa-B | Ego4D | 30.9 | 32.0 | 16.4 | 34.4 | 28.9 | 35.4 | 59.9 | 93.8 |
| LaViLa-B | Ego4D+HTM | 34.1 | 33.6 | 15.1 | 34.0 | 33.3 | 40.7 | 58.6 | 94.1 |
| LaViLa-B+EMBED | Ego4D+HTM | **36.0** | **34.9** | **19.0** | **39.0** | **37.0** | **42.7** | **61.3** | **94.5** |
| Helping Hands-L | Ego4D | 37.5 | **37.8** | - | - | 39.1 | 46.6 | 63.0 | 94.5 |
| LaViLa-L | Ego4D | 36.1 | 34.6 | 20.8 | 41.4 | 34.1 | 40.1 | 63.1 | 94.5 |
| LaViLa-L | Ego4D+HTM | 39.8 | 36.0 | 21.1 | 43.1 | 36.0 | 43.0 | 63.0 | **95.6** |
| LaViLa-L+EMBED | Ego4D+HTM | **40.8** | 37.5 | **22.8** | **45.0** | **40.3** | **46.7** | **64.7** | **95.6** |

Table 1: Zero-shot performance of models of different sizes (base 'B' and large 'L'). EMBED achieves the best performance compared with prior arts across tasks, including absolute gains of 4.6% on EK-100 MIR and 6.2% on EGTEA over LaViLa. The best scores are in **bold**.

where $s(x, y)$ represents the text-vision similarity score computed by a dot product between the model learned representations of $x$ and $y$, and $\tau$ is a temperature parameter that scales the similarity scores.

## 3 EXPERIMENTS

**Pretraining Datasets.** We pretrain models with both egocentric and exocentric datasets. For the egocentric data, we use the video-narration pairs from Ego4D following (Zhao et al., 2023; Li et al., 2021b). The resulting data consists of around 9K videos and 4M video-narration pairs in total. For the exocentric data, we use the HTM-AA dataset (Han et al., 2022) as the data source, which is a clean subset of the HowTo100M dataset (Miech et al., 2019b) that contains around 247K HowTo100M videos and 3.3M video-narration pairs.

**Baselines.** We apply EMBED to the LaViLa model (Zhao et al., 2023) due to its strong performance, and our primary comparisons are with 1) the original LaViLa model (Zhao et al., 2023), and 2) LaViLa fine-tuned using both the Ego4D and the original HTM-AA datasets. In addition, we present the performances of other vision-language pre-trained models, including EgoVLP (Li et al., 2021b), EgoVLPv2 (Pramanick et al., 2023), Xu et al. (2024), and Helping Hands (Zhang et al., 2023) for reference. We also adapt Ego-Exo (Li et al., 2021b) in video-language pretraining setting and compare with it in Appendix.

**Downstream Tasks.** We evaluate models on multiple egocentric downstream tasks as shown in Table 8. Specifically, we evaluate models on 1) Epic-Kitchens-100 (Damen et al., 2020) multi-instance retrieval (EK-100 MIR) and action recognition tasks; 2) Ego4D (Grauman et al., 2022) multiple choice questions (EgoMCQ) (Li et al., 2021b), and natural language query (EgoNLQ) and moment query (EgoMQ) tasks; 3) EGTEA (Li et al., 2018) action recognition that is focused on fine-grained cooking activities and 4) CharadesEgo (Sigurdsson et al., 2018) action recognition that classifies daily human indoor activities. We also experiment on HMDB-51 (Kuehne et al., 2011) and UCF-101 (Soomro et al., 2012) so as to assess the model performance on exocentric tasks.

**Evaluation Protocols.** We mainly focus on zero-shot evaluations where the pretrained video and text representations are directly utilized on the downstream video-text retrieval and action classification tasks, without any additional tuning specific to the downstream dataset. Following previous work (Li et al., 2021b; Zhao et al., 2023; Pramanick et al., 2023), we also report fine-tuning evaluations that involve adapting the pretrained video-text model through end-to-end fine-tuning using the training data of the target downstream dataset. Additionally, we evaluate models on exocentric tasks in the linear probing setting. In this setting, the pretrained video features are utilized as input, upon which a linear SVM is trained using the training subset of the downstream dataset.

| Rephraser | Narrator | Temporal | Spatial | EK-100 MIR | | EK-100 CLS | | EGTEA | | EgoMCQ | |
|---|---|---|---|---|---|---|---|---|---|---|---|
| | | | | mAP | nDCG | top-1 acc. | top-5 acc. | mean acc. | top acc. | intra acc. | inter acc. |
| | | | | 34.1 | 33.6 | 15.1 | 34.0 | 33.3 | 40.7 | 58.6 | 94.1 |
| ✓ | | | | 34.9 | 33.9 | 17.5 | 37.5 | 34.5 | 38.7 | 60.5 | 94.2 |
| | ✓ | | | 34.3 | 34.1 | 18.4 | 37.8 | 36.1 | 41.4 | 61.2 | 94.3 |
| | ✓ | ✓ | | 34.6 | 34.4 | 17.9 | 38.5 | 36.8 | 42.2 | **61.6** | 94.4 |
| ✓ | ✓ | ✓ | | 35.2 | 34.7 | 18.3 | 38.1 | 36.2 | 40.7 | 60.9 | **94.5** |
| ✓ | ✓ | ✓ | ✓ | **36.0** | **34.9** | **19.0** | **39.0** | **37.0** | **42.7** | 61.3 | **94.5** |

**Table 2:** Ablations on different modules of EMBED, including our narrator, rephraser, HOI clip temporal selection, and HOI region spatial focus techniques. Each of the techniques contributes to the model performance and combining them leads to the most robust performance.

| Model | Pretrain Data | EK-100 MIR | | EK-100 CLS | EGTEA | Charades-Ego | EgoNLQ | EgoMQ | |
|---|---|---|---|---|---|---|---|---|---|
| | | mAP | nDCG | top-1 acc. | mean acc. | mAP | R1@0.5 | R1@0.5 | mAP |
| EgoVLPv2-B | Ego4D | 47.3 | 61.9 | - | - | 34.1 | 7.9 | 31.1 | 12.2 |
| Helping Hands-L | Ego4D | - | - | - | - | - | 7.9 | 33.4 | **16.0** |
| LaViLa-L | Ego4D | 50.9 | 66.5 | 51.0 | 76.0 | 36.1 | 7.3 | 32.5 | 13.4 |
| LaViLa-L | Ego4D+HTM | 54.9 | 67.6 | 51.3 | **76.1** | 36.5 | 8.0 | 33.5 | 14.0 |
| LaViLa-L+EMBED | Ego4D+HTM | **56.0**$^*$ | **67.9**$^*$ | **51.9**$^*$ | **76.1** | **37.0**$^*$ | **8.5**$^*$ | **33.9**$^*$ | 15.1 |

**Table 3:** Fine-tuning performance of models of different sizes (base 'B' and large 'L'). EMBED outperforms baselines consistently in retrieval, classification, natural language query, and moment query tasks. $^*$ indicates significant improvements compared with the best baseline ($p < 0.05$ with paired bootstrap resampling).

**Implementation Details.** We train the LaViLa (Zhao et al., 2023) model on our constructed data. Llama-2-7B is used for narration paraphrase and the LaViLa-Narrator (Zhao et al., 2023) is used for narration generation, whose vision encoder is TimeSformer-Large and the text decoder is a GPT-2-XL (Radford et al., 2019). The hand-object interaction regions are pre-extracted with (Shan et al., 2020). We sample 4 frames with the resolution being $224 \times 224$ for each video clip during pretraining and 16 frames during finetuning. We initialize the models with the LaViLa parameters and train all the parameters jointly on Ego4D and HTM-AA for 5 epochs with the batch size set to 1024.

## 3.1 MAIN RESULTS

We compare our model, EMBED, with the baselines in zero-shot settings. As depicted in Table 1, directly training LaViLa with a combination of the Ego4D and HTM-AA datasets does not consistently enhance performance and can sometimes hinder it. For instance, LaViLa-B, trained on both Ego4D and HTM-AA, achieves a top-1 accuracy of 15.1% on the EK-100 CLS task and an intra-class accuracy of 58.6% on EgoMCQ, underperforming the original LaViLa-B model trained solely on Ego4D, which scores 16.4% and 58.6% respectively.

On the other hand, EMBED consistently outperforms the LaViLa baseline across various tasks. In the EK-100 MIR task, EMBED-B achieves mAP and nDCG scores of 36.0 and 34.9, surpassing LaViLa-B's 34.1 and 33.6. In the EK-100 CLS task, our model demonstrates robust performance with a top-1 accuracy of 19.0% and a top-5 accuracy of 39.0%, outperforming the baseline's 16.4% and 34.0%, respectively. Additionally, EMBED leads to significant gains in the EGTEA dataset, with mean accuracy reaching 37.0%, and in the EgoMCQ task, it yields superior intra-class performance of 61.3% and inter-class performance of 94.5%. Given that the primary difference between LaViLa-Ego4D+HTM and EMBED lies in the application of EMBED of the exocentric dataset, these results clearly emphasize the importance of dataset curation during joint training, as well as the effectiveness of our proposed EMBED method.

| Data | HMDB acc. | UCF acc. |
|---|---|---|
| Ego4D | 57.1 | 84.1 |
| HTM | 61.5 | 88.1 |
| Ego4D+HTM | 62.5 | 90.3 |
| Ego4D+HTM-EMBED | **63.8** | **90.7** |

**Table 4:** Evaluation results of LaViLa-L on exocentric tasks including HMDB-51 and UCF-101, measured in the linear probing setting. EMBED outperforms baselines trained on either egocentric, exocentric, or combined data sets.

| Model | EK MIR mAP | EK CLS top-1 acc. | EGTEA mean acc. | EgoMCQ intra acc. |
|---|---|---|---|---|
| *Ego4D+Kinetics-700* | | | | |
| LaViLa-B | 32.0 | 16.4 | 33.5 | 60.4 |
| LaViLa-B+EMBED | **33.3** | **17.0** | **34.5** | **61.4** |
| *Ego4D+COIN* | | | | |
| LaViLa-B | 30.9 | 15.8 | 26.0 | **60.7** |
| LaViLa-B+EMBED | **32.1** | **16.9** | **30.0** | **60.7** |
| *Ego4D+SSv2* | | | | |
| LaViLa-B | 31.0 | 15.3 | 33.2 | 60.6 |
| LaViLa-B+EMBED | **31.9** | **16.1** | **34.3** | **60.9** |

**Table 5:** Model performance when integrating Ego4D and different exocentric datasets, including Kinetics-700, COIN, and Something-Something v2. EMBED demonstrates consistent improvements over baselines when applied to various datasets.

Notably, EMBED surpasses previous models like EgoVLPv2 and Helping Hands in nearly all tasks within the zero-shot setting without sophisticated techniques such as hard negative sampling and EgoNCE (Li et al., 2021b), setting new state-of-the-art standards across various tasks.

## 3.2 ANALYSIS

**Ablations on Different Modules.** Table 2 illustrates the ablation study conducted to evaluate the individual contributions of different components within our proposed model, which includes our rephraser, narrator, HOI clip selection, and HOI region focus techniques. The findings highlight several key insights: 1) Unifying the language narration style enhances performance; 2) The narrator model proves effective, particularly when applied to the selected HOI clips rather than the original video clips; 3) The integration of EMBED in both language and video domains is beneficial, with their combined use markedly enhancing the model's capabilities in egocentric video understanding. Overall, each component positively impacts the model's performance, with the most substantial improvements observed when all components are utilized together.

**Finetuning Evaluation.** In the context of fine-tuning settings, Table 3 demonstrates how our EMBED model and other comparison models perform after fine-tuning on various datasets and tasks. We can see that EMBED achieves consistent improvements over baselines across tasks in the fine-tuning setting. For the EK-100 MIR task, EMBED achieves an mAP of 56.0, which is a notable improvement over LaViLa-Ego4D and LaViLa-Ego4D+HTM that score 50.9 and 54.9 respectively. For the classification tasks including EK-100 CLS, EGTEA, and Charades-Ego, EMBED retains its effectiveness and outperforms the baselines. EgoNLQ and EgoMQ are two relatively complicated tasks that require models to localize instances based on a language query or activity name. We follow previous works (Li et al., 2021b; Zhang et al., 2023) to finetune VSLNet (Zhang et al., 2020) with its input representations replaced with our pretrained representations. In both of the tasks, our model achieves competitive or superior performance compared with the baseline models, suggesting its robustness across settings.

**Exocentric Task Performance.** While our main focus is the model's egocentric understanding ability, here we also explore the model performance on exocentric tasks. We compare EMBED with the LaViLa baseline, each trained on varying datasets, using the HMDB-51 (Kuehne et al., 2011) and UCF-101 (Soomro et al., 2012) datasets for linear probing evaluation. As shown in Table 4, combining Ego4D and HTM-AA can improve the model performance on the exocentric tasks. However, EMBED can still maintain competitive or superior performance compared with the baseline models in this setting. This indicates that unifying egocentric and exocentric data in a unified format and mitigating their domain gap not only preserves but potentially enhances performance in exocentric settings, underscoring the versatility of our approach.

| Model | EK-100 CLS | | EGTEA | |
|---|---|---|---|---|
| | HOI acc. | non-HOI acc. | HOI acc. | non-HOI acc. |
| LaViLa-L | 15.0 | 16.1 | 33.5 | 17.7 |
| LaViLa-L+EMBED | 19.1 (+4.1) | 17.5 (+1.4) | 37.1 (+3.6) | 20.0 (+2.3) |

**Table 6:** Model performance on HOI and non-HOI instances. The improvements are more pronounced when HOI regions are detected, indicating that the improvements are mainly due to a better use of the HOI information.

**EMBED with Common Video Datasets.**    Our previous focus is on applying our model to Ego4D and HowTo100M. In this paragraph, we experiment with integrating Ego4D and other popular exocentric datasets, including Kinetics-700 (Carreira et al., 2019), COIN (Tang et al., 2019), and Something-Something v2 (Goyal et al., 2017). Table 5 shows that EMBED demonstrates robust performance across these varied datasets, indicating its adaptability and effectiveness in diverse contexts. We refer readers to Appendix for the details.

**Performance on HOI and non-HOI instances.**    Our method mainly uses HOI information to perform the dataset alignment. In this part, we analyze if the model trained on our dataset can effectively utilize the HOI information. To this end, we split evaluation sets into two groups: HOI instances and non-HOI instances, depending on if there are HOI regions detectable by an HOI detector (Shan et al., 2020). As shown in Table 6, our improvements over baselines are much more pronounced when there are HOI regions detected, indicating that the improvements of the model are mainly because of the better use of HOI information. This verifies that focusing on HOI information is an effective way to improve the model egocentric performance.

| Model | Pretrain Data | EK-100 MIR | | EK-100 CLS | |
|---|---|---|---|---|---|
| | | top-1 acc. | top-5 acc. | mean acc. | top acc. |
| LaViLa-B | HTM-AA | 22.2 | 26.4 | 3.5 | 14.0 |
| LaViLa-B+EMBED | HTM-AA | 25.9 | 28.8 | 11.2 | 28.6 |

**Table 7:** Results on pretraining with HTM-AA only. Our method can still improve the model performance when the model is trained with only exocentric data.

**Experiments with HTM-AA only.**    Previously, we pretrain models with both Ego4D and HTM-AA datasets. In this part, we investigate how the models will perform if only trained on the HTM-AA dataset. As shown in Table 7, in this setting, EMBED can still improve LaViLa by a big margin thanks to better utilization of exocentric data for egocentric learning.

## 4    RELATED WORK

**Egocentric Video Understanding.**    Understanding videos from an egocentric perspective introduces unique research challenges, including areas like action recognition (Sigurdsson et al., 2018) and hand/body pose estimation (Ohkawa et al., 2023; Jiang & Grauman, 2017). Nevertheless, egocentric datasets have historically been small and specialized, which has held back research on egocentric video learning. Notably, many works initialize their models with parameters trained on exocentric data (Zhao et al., 2023), due to the scarcity of relevant egocentric data. However, the surge in the size of egocentric video datasets (Damen et al., 2018; 2020; Grauman et al., 2022; 2024) over recent years has brought about fresh opportunities and complexities (Li et al., 2021b; Pramanick et al., 2023). For example, Ego-Only (Wang et al., 2023) shows that egocentric video representation can now be trained with egocentric data only, without transferring from exocentric videos or images. In contrast, our research aims to find new ways to make exocentric data useful for better understanding egocentric videos in this evolving context.

**Vision-Language Pretraining.**    Vision-language (VL) pretraining has first demonstrated effective for image representation learning (Lu et al., 2019; Tan & Bansal, 2019; Su et al., 2019; Li et al.,

2019; Chen et al., 2020). These models, when presented with both visual and textual inputs, encode them either separately (Radford et al., 2021; Jia et al., 2021) or in a joint manner (Kim et al., 2021; Li et al., 2021a; Dou et al., 2022). Subsequently, they are trained to align the representations of corresponding vision-language pairs through contrastive (e.g. InfoNCE (van den Oord et al., 2018)) and/or image-conditioned language modeling losses (e.g. masked language modeling (Devlin et al., 2019)). The advent of large-scale video-language datasets (Bain et al., 2021; Carreira et al., 2019; Krishna et al., 2017; Miech et al., 2019b) has facilitated the extension of similar pretraining methodologies into the realm of videos (Sun et al., 2019b;a; Li et al., 2020). However, due to the inherent difficulty in gathering high-quality video-language data, researchers have made efforts to adapt existing approaches to handle noisy video-language datasets (Miech et al., 2019a). In contrast to many uncurated video datasets, Ego4D (Grauman et al., 2022) stands out as a collection of high-quality videos meticulously annotated with timestamped language narrations. This resource has spurred the development of numerous pretraining models for video-language tasks (Li et al., 2021b; Pramanick et al., 2023; Zhao et al., 2023; Zhang et al., 2023; Ashutosh et al., 2023). Yet, most of these models have predominantly focused on videos captured from either egocentric or exocentric perspectives alone. For example, LaViLa (Zhao et al., 2023) focuses on data augmentation within egocentric sources. Consequently, the challenge of effectively combining datasets featuring different viewpoints for video-language training remains relatively underexplored. In contrast, our approach introduces methods for automatically mining video clips from exocentric sources and we demonstrate significant empirical improvements over previous models such as LaViLa across settings.

Recently, there are works on using language models to re-write or re-generate narrations for videos (Shvetsova et al., 2023; Xu et al., 2024). For example, Xu et al. (2024) propose to retrieve relevant exocentric data for egocentric videos and re-generate better narrations for pretraining. In contrast to this line of work, our approach do not rely on existing video clips, but instead actively mines new video clips and pairs them with their corresponding language narrations. In addition, our focus in on adapting data from one viewpoint to another rather than performing data cleaning.

**Cross-View Video Learning.** There has been prior work aimed at bridging the gap between egocentric and exocentric video perspectives (Sigurdsson et al., 2018; Ardeshir & Borji, 2018; Joo et al., 2015; Fan et al., 2017; Yonetani et al., 2015). One prevalent strategy involves the development of viewpoint-invariant representations through embedding learning techniques, which have been applied in domains such as action recognition (Soran et al., 2015) and person segmentation (Xu et al., 2018). Another line of research focuses on image generation methods that employ generative adversarial frameworks to synthesize one viewpoint from the other (Regmi & Borji, 2018; Regmi & Shah, 2019). Additionally, some efforts have treated viewpoint invariance as a domain adaptation task, adapting exocentric video models for overhead drone-footage scenarios (Choi et al., 2020). However, most of these approaches require paired datasets (Grauman et al., 2024; Huang et al., 2024), either simultaneously recorded or sharing the same labels, across different viewpoints. Ego-Exo (Li et al., 2021b) eliminates the need for videos concurrently recorded from both viewpoints by identifying latent egocentric signals in exocentric video classification data and distilling this knowledge into the video encoder. Unlike Ego-Exo, we are focused on video data with flexible language narrations, which extends beyond categorical labels and has wider and more flexible applicability. In addition, we proactively select videos rich in egocentric cues to streamline the identification and learning of hand-object interactions, curating new video-language datasets with a large number of videos tailored for egocentric learning, whereas Ego-Exo passively utilizes existing video data. In addition, we also empirically demonstrate that incorporating Ego-Exo objectives cannot improve video-language pretraining in Appendix, confirming the necessity of proposing new methods in this direction.

## 5 CONCLUSION

EMBED improves egocentric video understanding by unlocking the untapped potential of exocentric video data. By identifying egocentric cues from exocentric data, we actively search for egocentric-relevant video clips and pair them with action-focused language narrations, resulting in exocentric-sourced data tailored for egocentric video-language pretraining. The extensive evaluations of EMBED demonstrate its strong performance, achieving significant improvements over strong baselines on multiple benchmarks. Our findings encourage further exploration into the combination of egocentric and exocentric perspectives in video understanding and beyond.

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

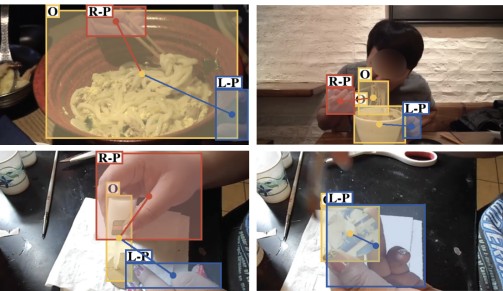

**Figure 3:** The HOI detector can accurately extract the right hand (R-P), left hand (L-P) and object (O) regions from a video frame.

Hao Zhang, Aixin Sun, Wei Jing, and Joey Tianyi Zhou. Span-based localizing network for natural language video localization. In *ACL*, 2020.

Yue Zhao, Ishan Misra, Philipp Krähenbühl, and Rohit Girdhar. Learning video representations from large language models. In *CVPR*, 2023.

## A    PRETRAINING DETAILS

In this section, we go over the implementation details of each of our modules during the pretraining stage.

**HOI Detector.**    We use the hand-object detection model (Shan et al., 2020) trained on the 100-DOH dataset.[2] We extract the hand and object regions using the off-the-shelf model from the video frames. Figure 3 demonstrates that the HOI detector can achieve robust performance in the exocentric setting.

**HOI Video Clip Selection.**    As mentioned in the main paper, we perform data selection to select video clips capturing hand-object interactions. Specifically, we segment all the videos in the HTM-AA dataset into 5-second video clips. For each of the video clips, we uniformly sample 4 video frames from it and compute the HOI score accordingly. The video clips with the highest HOI scores are selected for training.

Figure 4 showcases video clips of the highest and lowest HOI scores respectively. We can see that our data selection strategy can select video clips that capture close-up hand-object interactions that are akin to the egocentric dataset.

**Spatial Focus.**    As shown in Figure 5, given a video clip consisting of several frames, we use the HOI detector to detect the hand and object regions from all the frames. Then, we take the convex hull of the extracted regions so that it can cover all the hand and object regions in this video. Finally, we crop this region out of the video frame and feed this cropped input to the model.

**Exo-to-Ego Rephraser.**    Many of the exocentric narrations are redundant and contain information irrelevant to human actions. To filter these narrations, we first train a text classification model on the HTM-Align dataset (Han et al., 2022) to classify whether a sentence is useful or not. HTM-Align is a manually annotated 80-video subset of HowTo100M. It is randomly sampled from the Food and Entertaining category of HowTo100M. Each sentence is annotated with whether it is visually alignable with the video and its corresponding start and end timestamps within the video. Given the HTM-Align dataset, we finetune the DeBERTa-v3-base (He et al., 2021) model on it for this binary prediction task. Specifically, we append a classification layer on top of the pretrained DeBERTa-v3 and fine-tune the whole model for 3 epochs with the learning rate set to 5e-5. The trained checkpoint is then used for filtering sentences that are classified as non-visually alignable. Note that DeBERTa-v3 is a text-only model that does not take any vision inputs.

---

[2]https://github.com/ddshan/hand_detector.d2

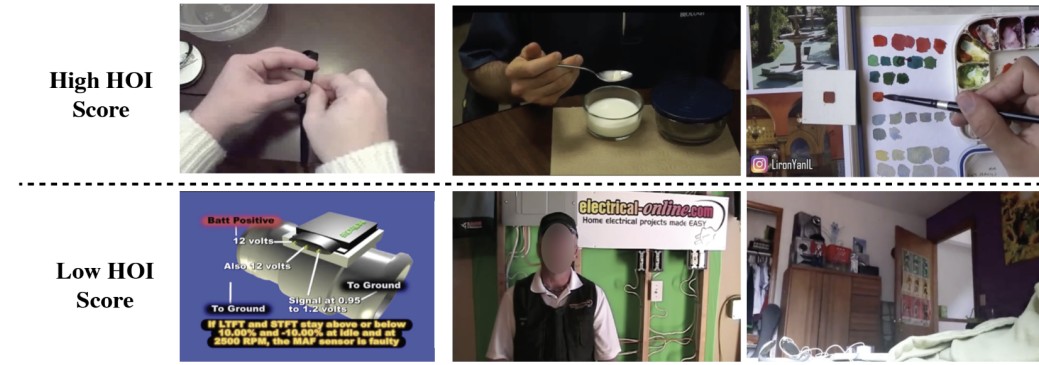

**Figure 4:** Video clips with high and low HOI scores. Videos with high HOI scores typically contain close-up hand-object interactions whereas videos with low HOI scores do not capture any human actions.

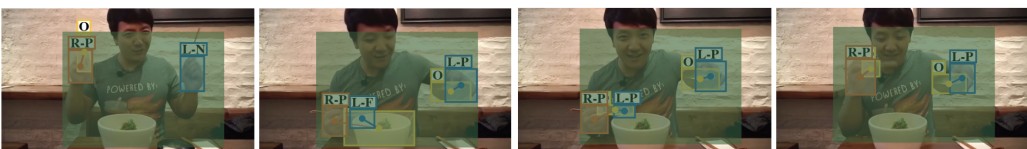

**Figure 5:** Demonstrate of HOI region spatial focus. Given a video clip, we extract the hand (in red and blue) and object (in orange) regions from each frame. We then compute the convex hull of all the boxes (in green) and crop the regions.

After filtering the non-visually-alignable sentences, we then use the Llama-2 model (Touvron et al., 2023) to perform exo-to-ego rephrasing. The Llama-2 model is pretrained with 2 trillion tokens and is then finetuned for chat use cases. Its code and model weights are publicized,[3] and we use the Llama-2-7B-Chat model without any finetuning. Similar to DeBERTa-v3, Llama-2 does not take vision inputs, thus it performs paraphrasing given text inputs only.

To use the Llama-2 model for our purpose, we provide it with the instruction and several input-output examples as shown below:

```
## Instruction
System: You are an assistant that extracts actions given the user inputs.

## Exo-to-Ego Rephrasing Examples
## User: Input; Assistant: Output.
User: and finally i'll route the rest of the hair here
Assistant: route the rest of the hair

User: the clay is pressed into shape over the mold
Assistant: press the clay into shape over the mold

User: let's start by turning on my stove
Assistant: turn on the stove

...

## Rephrasing New User Input
User: <Input>
```

To illustrate, we first instruct the model that they are an assistant that extracts actions given the user inputs. Then, we provide several pairs of exo-to-ego narration translation examples that further demonstrate how the model should perform the translation. In this way, the Llama-2 model is able to

---

[3]https://github.com/facebookresearch/llama

| Datasets | Task | Metrics |
|---|---|---|
| EK-100 (Damen et al., 2020) | MIR | mAP, nDCG |
| EK-100 (Damen et al., 2020) | CLS | action acc. |
| Ego4D (Grauman et al., 2022; Li et al., 2021b) | MCQ | inter-/intra-video acc. |
| Ego4D (Grauman et al., 2022) | NLQ, MQ | recall |
| EGTEA (Li et al., 2018) | CLS | top-1, mean acc. |
| CharadesEgo (Sigurdsson et al., 2018) | CLS | mAP |
| HMDB-51 (Kuehne et al., 2011) | CLS | mean acc. |
| UCF-101 (Soomro et al., 2012) | CLS | mean acc. |

**Table 8:** Our evaluation of EMBED includes a diverse range of tasks across several datasets. We assess its performance on the Epic-Kitchens-100 (EK-100) dataset for both multi-instance retrieval (MIR) and action recognition (CLS) tasks, the Ego4D dataset for multiple-choice question (MCQ), natural language query (NLQ), and moment query (MQ) tasks, and also the EGTEA and CharadesEgo datasets for action recognition (CLS) tasks. We also experiment on exocentric tasks, including HMDB-51 and UCF101. We refer readers to Appendix for more details.

perform the exo-to-ego rephrasing given any new user inputs and we use this model to rephrase all the exocentric narrations.

**Ego Narrator.** We use the narrator model trained on the Ego4D dataset by LaViLa (Zhao et al., 2023). Specifically, the LaViLa model trains a narrator model consisting of a TimeSFormer vision encoder and a GPT-2 language decoder on the Ego4D dataset and then uses it to perform inference on Ego4D to enrich the original egocentric dataset. Here, we repurpose this model for generating egocentric-style narrations given exocentric videos. We use beam sampling with the beam size set to 5 when generating the narrations.

**Ego4D.** Ego4D contains 3,670 hours of egocentric videos that are densely annotated with language narrations. Each narration is a free-form text sentence and is annotated with its timestamp within the video. We follow previous work (Zhao et al., 2023; Li et al., 2021b) to prepare the Ego4D dataset for vision-language pretraining. Specifically, videos that appear in the validation and test sets of Ego4D are excluded and each language narration corresponds to a video clip with its start and end timestamps determined by a heuristic (Li et al., 2021b). Also, narrations that contain the "#unsure"/"#Unsure" tags or are shorted than 4 words are removed. The resulting dataset consists of about 8K videos and 4M video-text pairs with an average clip length of about 1 second. Note that we do not use the LaViLa augmented dataset in this paper.

**HTM-AA.** HTM-AA means the Auto-Aligned (AA) version of HowTo100M and it is a clean subset providing matched video-text pairs (Han et al., 2022). We use the HTM-AA version-1 which consists of about 250k HTM videos and 3M video-text pairs. Similar to Ego4D, each narration $l$ is annotated with a timestamp $t$ and we pair each narration with its corresponding 5-second video clip $[t - 2.5, t + 2.5]$.

**Training.** We use the publicized LaViLa codebase for training the baseline and our model.[4] Each model is initialized with the LaViLa model pretrained on their augmented Ego4D dataset and is then finetuned on both Ego4D and HTM with a fixed learning rate of 1e-5 for 5 epochs. We scale the short side of both the Ego4D and HTM videos to 288 pixels to reduce storage and accelerate training. We uniformly sample 4 video frames from each video clip and resize the frames to 224x224.

## B EVALUATION DETAILS

**EK-100.** Epic-Kitchens-100 contains 100 hours of egocentric cooking videos. The training/validation/testing splits of EK-100 consist of 67,217/9,668/13,092 video clips respectively. Each video clip is paired with its start and end timestamps, a language narration, as well as its corresponding verb and noun class.

---

[4]https://github.com/facebookresearch/LaViLa

| Sampling Strategy | EK-100 MIR | | EK-100 CLS | | EGTEA | | EgoMCQ | |
|---|---|---|---|---|---|---|---|---|
| | mAP | nDCG | top-1 acc. | top-5 acc. | mean acc. | top acc. | intraacc. | interacc. |
| Beam Sampling | 33.0 | 33.4 | 18.4 | 37.8 | 36.1 | 41.4 | 61.2 | 94.3 |
| Multinomial Sampling | 32.6 | 33.4 | 16.5 | 36.0 | 35.6 | 39.3 | 61.1 | 94.6 |

**Table 9:** Comaprisons of different sampling strategies. Beam sampling is better than multinomial sampling when generating narrations on the HTM dataset using the Ego4D-trained narrator.

| Exocentric Data Size | EK-100 MIR | | EK-100 CLS | | HMDB-51 | UCF-101 |
|---|---|---|---|---|---|---|
| | mAP | nDCG | top-1 acc. | top-5 acc. | acc. | acc. |
| 0.5M | 33.9 | 33.7 | 17.8 | 36.9 | 51.9 | 79.5 |
| 1M | 34.0 | 33.7 | 18.0 | 37.8 | 53.1 | 80.5 |
| 1.5M | 34.6 | 34.4 | 17.9 | 38.5 | 53.8 | 81.5 |

**Table 10:** Training models with different amounts of HOI score-selected exocentric data. Only the narrator model of EMBED is used on the HOI video clips in this setting.

| Exocentric Data Size | EK-100 MIR | | EK-100 CLS | | HMDB-51 | UCF-101 |
|---|---|---|---|---|---|---|
| | mAP | nDCG | top-1 acc. | top-5 acc. | acc. | acc. |
| 0.5M | 33.2 | 33.8 | 15.3 | 34.3 | 51.1 | 80.6 |
| 1M | 33.4 | 33.4 | 15.5 | 34.4 | 51.6 | 82.2 |
| 1.5M | 34.3 | 33.8 | 15.3 | 34.3 | 53.9 | 82.6 |

**Table 11:** Training baselines with different amounts of exocentric data randomly sampled from the original HTM dataset.

In the zero-shot setting, we evaluate the models on the EK-100 MIR and CLS validation set without any finetuning. Different from the pretraining stage, we sample 16 frames from each video clip instead of 4 frames so that the model can get more fine-grained information.

In the fine-tuning setting, to train the models, we use the language narration for EK-100 MIR and the verb/noun/action class label for EK-100 CLS. For EK-100 CLS, the evaluation metrics are top-1/5 action accuracies in the zero-shot setting and top-1 accuracies for verb, noun, and action in the fine-tuning setting, and the action accuracy is the most important evaluation metric. We follow LaViLa to set the hyper-parameters.

**EgoMCQ.** EgoMCQ is a multiple-choice question-answering dataset built on top of Ego4D (Li et al., 2021b), consisting of around 39K questions. The task is to match a narration to its corresponding video clip given 5 candidates sampled from either the same video ('intra-video') or other videos ('inter-video'). The dataset is focused on zero-shot evaluation and we report both the intra-video and inter-video accuracies.

**EgoNLQ/EgoMQ.** EgoNLQ and EgoMQ are two downstream tasks provided in the Ego4D benchmark. The task is to localize the temporal window within the video given a natural language query (EgoNLQ) or an activity name (EgoMQ). Following previous work (Li et al., 2021b; Zhang et al., 2023), we extract the video and text features with our pretrained models and feed them to VSLNet (Zhang et al., 2020) for fine-tuning. We report the top-1 accuracies with the ground truth at an IoU threshold of 0.5.

**EGTEA.** EGTEA is an egocentric cooking dataset consisting of 28 hours of cooking videos with gazing tracking. Its action annotations include 10,321 instances of fine-grained actions from 106 classes. We evaluate the pretrained model on all three splits of its test set and report the top-1 accuracy and mean-class accuracy. For fine-tuning, we follow LaViLa to set the hyper-parameters.

**CharadesEgo.** CharadesEgo is a dataset that contains both egocentric and exocentric videos. Different from other egocentric datasets, it mainly captures daily indoor activities. We use the

egocentric subset only, consisting of around 3K and 1K videos for training and testing respectively. We report the video-level mAP scores and follow LaViLa to set the hyper-parameters.

**HMDB-51/UCF-101.** HMDB-51 and UCF-101 are two exocentric video classification datasets. Following LaViLa, in the linear-probing evaluation process, the video encoder is frozen. We extract video features and train a linear SVM using these features. This process is applied to video clips from the HMDB-51 or UCF-101 datasets. We divide each video into four 32-frame clips, evenly sampled throughout the video. The video clips are fed to the video encoder to produce the final visual embeddings. For evaluation, we calculate the average prediction score across the different splits. The performance is measured using scikit-learn's LinearSVC, with the best top-1 accuracy determined by varying the regularization parameter C within the range of $\{10^{-5}, 10^{-4}, 10^{-3}, 10^{-2}, 0.1, 1, 10^2, 10^3, 10^4\}$.

**EMBED with Common Video Datasets.** In the main paper, we experiment with the integration of Ego4D with other well-known datasets using EMBED. Specifically, for the Kinetics-700 dataset, we adapt its original labels into language narrations. For instance, the label "clay pottery making" is rephrased to "a person is making clay pottery." In the case of the COIN and SSv2 datasets, we utilize their existing manually annotated language narrations, which are notably precise. Consequently, for these three datasets, we forego our rephraser and only apply our narrator, HOI clip selection, and spatial focus techniques. All other experimental parameters remain consistent with those used in the integration of Ego4D and HowTo100M.

## C  ADDITIONAL EXPERIMENTS

**Sampling Strategies.** LaViLa (Zhao et al., 2023) has previously demonstrated that the nucleus sampling works much better than beam search possibly because nucleus sampling introduces more diversity into its generations albeit at a cost of quality. We compare beam sampling with nucleus sampling in our paper, and as shown in Table 9, beam sampling is better than nucleus sampling. This is because when using the narrator in the out-of-domain setting, the generations are relatively of low quality and it is important to first ensure the generation quality when using the egocentrically-trained narrator in the exocentric setting.

**Comparisons with Ego-Exo (Li et al., 2021b).** Ego-Exo and our method share similar goals, as both approaches aim to utilize exocentric data for egocentric representation learning. However, Ego-Exo is focused on video classification data with categorical labels, making it hard to adapt it for general video-language pretraining settings where flexible language narrations are involved. To demonstrate this, we try to incorporate Ego-Exo into our video-language pretraining step by asking the video encoder to localize the HOI heatmaps detected by (Shan et al., 2020) using the objectives proposed in Ego-Exo (Li et al., 2021b). As shown in Table 12, incorporating Ego-Exo cannot improve the model performance in the video-language pretraining setting, suggesting its incompatibility with the current state-of-the-art paradigm.

| Model | Pretrain Data | EK-100 MIR | | EK-100 CLS | |
| --- | --- | --- | --- | --- | --- |
| | | top-1 acc. | top-5 acc. | mean acc. | top acc. |
| LaViLa-B w/ Ego-Exo +EMBED | Ego4D+HTM-AA | 35.0 | 34.3 | 18.8 | 38.8 |
| LaViLa-B+EMBED | Ego4D+HTM-AA | 36.0 | 34.9 | 19.0 | 39.0 |

Table 12: Incorporating the Ego-Exo objectives (Li et al., 2021b) cannot improve the model performance in the video-language pretraining setting.

**Scaling.** In this part, we train models with different amounts of exocentric data for both the baseline and our models. As we can see from Table 11, the model performance improves with an increasing amount of data and our method outperforms the baseline model on egocentric tasks. Furthermore, in line with our expectations, we observe that an increase in the amount of exocentric data correlates with enhanced performance in exocentric tasks.

**Applications in Other Models.**    Our constructed data can technically be used to train any video-language models. In the main content, we train the LaViLa model on our data. In this part, we investigate how other models perform when trained with our data. As shown in Table 13, we can see improvements over the Helping Hands model (Zhang et al., 2023) using our data, demonstrating that our method is compatible with other models as well.

| Model | Pretrain Data | EK-100 MIR | | EK-100 CLS | |
|---|---|---|---|---|---|
| | | top-1 acc. | top-5 acc. | mean acc. | top acc. |
| Helping Hands | Ego4D+HTM-AA | 38.5 | 36.2 | 25.1 | 45.9 |
| Heling Hands+EMBED | Ego4D+HTM-AA | 39.4 | 36.9 | 25.6 | 46.6 |

**Table 13:** Our method can be applied to models other than LaViLa and achieves improvements.

**Qualitative Results.**    We showcase the qualitative outcomes of our exo-to-ego rephraser and ego narrator in Figure 6. Both models demonstrate commendable performance. The rephraser and narrator, drawing from language and video inputs respectively, exhibit distinct characteristics. For instance, the rephraser effectively leverages the original narration to capture human actions, occasionally producing narrations that are more precise than those of the narrator (compare "cut the wingtip off" and "hold the chicken with both hands"). Conversely, the narrator proves more advantageous when the original narration does not align well with the video content (compare "give the whole model a wash using citadel's irqa's earth shade" and "dip the brush in the paint"). We utilize this complementary functionality of the two models and integrate them together into our method.

| Video | Narration | Rephraser | Narrator |
|-------|-----------|-----------|----------|
|  | when you have the rice mixing with the olive oil | mix the rice with the olive oil | stir the rice in the pot with the wooden spoon |
|  | now with our mixture we are going to pour it right over the meat | pour the mixture over the meat | pour the sauce on the meat |
|  | turn your pan on high and put 3 tablespoons of olive oil in it | turn on high heat and add 3 tablespoons of olive oil | pour oil in the pan |
|  | start cutting the pomelo first like that into and breaking it up into the segments and with the help of a little knife so you literally to scoop out the flesh and this really does gonna give the whole new meaning to christmas | start cutting the pomelo and breaking it up into segments | cut the bread with the knife |
|  | now we've cut the wingtip off now | cut the wingtip off | hold the chicken with both hands |
|  | the next step is to give the whole model a wash using citadels iraq's earth shade | give the whole model a wash using citadel's iraq's earth shade | dip the brush in the paint |

**Figure 6:** Qualitative results of our exo-to-ego rephraser and ego narrator. The rephraser effectively leverages the original narration to capture human actions, occasionally producing narrations that are more precise than those of the narrator (compare "cut the wingtip off" and "hold the chicken with both hands"). On the other hand, the narrator is more advantageous when the original narration does not align well with the video content (compare "give the whole model a wash using citadel's irqa's earth shade" and "dip the brush in the paint").

