# OpenReview forum: "Unlocking Exocentric Video-Language Data for Egocentric Video Representation Learning"
_ICLR.cc/2025/Conference — Submitted to ICLR 2025_

### Official Review · Reviewer_XASQ · 2024-10-24

**Soundness:** 3
**Presentation:** 3
**Contribution:** 2
**Rating:** 6
**Confidence:** 4

**Summary:**

This paper proposes EMBED, a data generation framework that leverages exocentric datasets for egocentric video-language pretraining. Unlike previous works that primarily learn from paired ego-exo data or unpaired full-scale exocentric videos (e.g., Ego-Exo, EgoExoNCE), EMBED focuses on learning HOI-related representations from zoomed-in, HOI-specific exocentric videos and egocentric-style text descriptions. The authors conduct comprehensive experiments across various egocentric and some simple exocentric benchmarks that demonstrates promising results.

**Strengths:**

- **Comprehensive Experimental Evaluation of EMBED**: I appreciate the authors' efforts in conducting diverse and thorough evaluations. They primarily validate their exocentric video-text pairs from the HTM-AA dataset across a wide range of commonly used egocentric benchmarks, as well as simpler exocentric benchmarks like HMDB and UCF101. They also attempt to demonstrate the effectiveness of EMBED using other third-person datasets like Kinetics-700, though this yields only minor improvements.
- **Promising Experimental Results**: EMBED improves LaViLa-B and LaViLa-L when continue to pretrain on Ego4D and HTM, offering a promising approach for egocentric video understanding by using filtered exocentric data (particularly HTM-AA) that highlights hand-object interactions.

**Weaknesses:**

- **Limited Technical Contribution:** The tools (HOI Detector, Narrator), data (Howto100M), and architecture (LaViLa) are mostly well-known in the egocentric domain. From prior works like Ego-Exo[CVPR2021], EgoVideo [CVPR 2024 EgoVis challenge champions] and EgoExoNCE [CVPR 2024], I've seen that incorporating exocentric data can offer some benefits. The approach of extracting HOI regions for training is somewhat similar to GroupRandomSizedCrop (if the crop just happens in HOI region and zoomed-in), a useful augmentation in the egocentric domain. Additionally, rephrasing exocentric text into egocentric text is an obvious and naive way to address the text domain gap. Overall, I don’t find much novelty here.
- **Concerns About Using the Exocentric Dataset HTM-AA:** While I appreciate the extensive experiments, the improvements seem to stem primarily from using the clean version of the HowTo100M dataset (HTM-AA). HTM-AA offers cleaner data and includes a significant amount of kitchen activities, which overlap with scenarios in Epic-Kitchens and EGTEA, likely contributing to the larger improvements in these downstream tasks. Table 5 further shows that EMBED gains little and incurs high costs when pretraining with additional noisy datasets like K700, strenghthening my concerns about the overall effectiveness of EMBED.
- **Concerns About Initialization with LaViLa:** I have concerns of the approach to initialize with LaViLa (mentioned in Lines 356-357) rather than pretraining from scratch when fine-tuning all parameters. By initializing with LaViLa, the overall computational cost—combining the training of LaViLa on Ego4D and fine-tuning on Ego4D+HTM—becomes twice that of LaViLa alone, which raises concerns about a potentially unfair comparison in Table 1.

**Questions:**

- **About Selecting Visually Aligned Sentences:** Why is the selection conducted by classifying the text alone? Wouldn't the mismatch between video and sentences be determined by both the video and the text? It seems that the classification focuses on distinguishing HOI from non-HOI sentences. Please correct me if I’m wrong.
- **Limited Improvement with SSv2:** Despite SSv2 being more similar to egocentric visual content, the improvement is even smaller than with the noisier K700 dataset. I'm curious about the underlying reasons for this.
- **Obvious Typo in Abstract, Line 20-21**: ""..we construct datasets thar are..."", thar-->that.

---

> ### Author Response · Authors · 2024-11-23
> **Response**
>
> > Limited Technical Contribution
>
> We acknowledged existing works that leverage exocentric datasets for egocentric learning in our paper and provided comparisons in both the experiment and the related work sections. Additionally, we highlight our contributions in the general response.
>
> Our proposed techniques are simple yet effective, demonstrating clear improvements by leveraging exocentric datasets compared with our baselines. We believe these contributions should not be dismissed as obvious and naive but recognized for their practicality. Simplicity and intuitiveness should be viewed as strengths, not weaknesses.
>
>
> > Concerns About Using the Exocentric Dataset HTM-AA
>
> We would like to emphasize that our main baseline models are trained using the exact same data sources. For instance, in Table 1, the main LaViLa baselines are trained with Ego4D and the original HTM-AA dataset. As a result, the improvements achieved by our method over the baselines cannot be attributed to domain similarity between the training and evaluation data.
>
> Furthermore, we conducted additional experiments on egocentric QA tasks and embodied tasks, as detailed in our response to Reviewer rhcR, further demonstrating the effectiveness of our method across different domains.
>
>
>
> > Concerns About Initialization with LaViLa
>
> We would like to stress that our main baseline models are also initialized from LaViLa. Specifically, both our baseline and EMBED are first initialized with LaViLa and then pretrained on Ego4D and either the original HTM-AA dataset or our constructed version. Furthermore, both the original HTM-AA dataset and our constructed dataset are approximately the same size, ensuring the fairness of our comparisons.
>
> In summary, our experimental setting is strictly controlled, with the only difference between our LaViLa-Ego4D+HTM-AA baseline and EMBED being the use of either the original HTM-AA dataset or our constructed version.
>
> > Selecting Visually Aligned Sentences
>
> Thank you for your question! Our preliminary findings indicate that non-visually alignable sentences often lack action-related information, which can typically be identified based on text alone. We found that this simple strategy is effective, and we will revise the wording to make this clearer.
>
>
> > Limited Improvement with SSv2
>
> For the SSv2 experiments, the baseline is trained on the original SSv2 dataset, while our model is trained on our constructed version. Since the original SSv2 dataset already shares similarities with egocentric visual content, it is expected that it is less necessary to use our framework and thus we would yield relatively smaller improvements in this case.
>
>
> > Typo
>
> Thank you for catching the typo! We will proofread the draft carefully in the revised version.

---

> > ### Comment · Reviewer_XASQ · 2024-11-26
> >
> > I would like to thank the authors for their efforts during the rebuttal process. I have reviewed the authors' rebuttal, general response, and the other reviews. Below is my summary after rebuttal:
> > - The Major Concern Remains: My primary concern remains the limited technical contribution. The insights presented--training on third-person datasets with a zoom-in cropping technique to improve egocentric video understanding--feel somewhat incremental within the egocentric literature.
> > - The Secondary Concern Remains: The improvements largely stem from HTM-AA, which offers cleaner hand-object interaction data and contain a massive number of kitchen activities. This heavy reliance on HTM-AA, rather than the proposed EMBED approach, might limit the novelty and broader applicability of the method.
> >
> > Considering the authors’ significant efforts during the rebuttal, I am inclined to raise my rating from borderline reject to borderline accept.
> >
> > Lastly, I remain unsure whether the contributions and insights presented in this paper sufficiently meet the ICLR standard.

---

> > > ### Author Response · Authors · 2024-12-02
> > > **Thank you!**
> > >
> > > We sincerely thank the reviewer for their thoughtful comments and for raising their rating. Regarding the remaining concerns, we would like to state that
> > >
> > > - We systematically identify the challenges of leveraging exocentric data for egocentric tasks and propose practical, simple, and effective solutions. The significant improvements achieved over robust baselines, including standard action recognition benchmarks and challenging QA and embodied tasks, validate the effectiveness and value of our framework.
> > >
> > > - EMBED's improvements cannot be attributed solely to HTM-AA. Both baseline and EMBED use identical data sources and initialization. Controlled experiments and additional evaluations on diverse tasks confirm the robustness and broader applicability of our method.
> > >
> > > We appreciate the reviewer’s acknowledgment of our paper’s strengths. We believe our work meaningfully contributes to the field of egocentric video understanding and meets the ICLR standard, addressing critical challenges with effective techniques.

---

### Official Review · Reviewer_baF8 · 2024-10-28

**Soundness:** 3
**Presentation:** 3
**Contribution:** 3
**Rating:** 6
**Confidence:** 5

**Summary:**

The paper presents EMBED, a framework for enhancing egocentric video representation learning using exocentric video-language data. The method involves: 1.Video Clip Curation: Selecting video clips from exocentric datasets that emphasize hand-object interactions and refining them spatially to focus on these regions. 3.Language Narration Generation: Using an exo-to-ego rephraser to transform existing narrations and an ego narrator to generate new ones in egocentric style. The experimental results indicate that this method achieves performance improvements across egocentric tasks.

**Strengths:**

1.This method has achieved a significant performance improvement across several tasks.
2.The authors conduct thorough ablation experiments.
3.The authors propose a framework that utilizes a large amount of third-person video data to assist in understanding egocentric video data, which is significant for advancing first-person video understanding.

**Weaknesses:**

1. It is unclear whether the authors intend to release their models and datasets.
2. There have already been some works on refining annotations using LLMs and leveraging exocentric videos to assist with egocentric videos, making the entire framework not novel.
3. How is the quality of annotations on exocentric videos using an HOI detector and how do the authors handle erroneous data?

**Questions:**

See weaknesses above.

---

> ### Author Response · Authors · 2024-11-23
> **Response**
>
> > It is unclear whether the authors intend to release their models and datasets.
>
> We plan to release them upon paper publication.
>
> > There have already been some works on refining annotations using LLMs and leveraging exocentric videos to assist with egocentric videos, making the entire framework not novel.
>
> As in our general response, our framework 1) actively mines egocentric-relevant data from exocentric sources, which extends beyond simply refining existing annotations; 2) leverages exocentric data with flexible language annotations in a relatively large-scale setting, outperforming existing works with a similar motivation such as Ego-Exo.
>
> Considering these points, we view our approach not as a data refining process but as an active video mining and tailored language generation process that has not been explored before. Also, our method is more generally applicable and achieves better performance than existing methods such as Ego-Exo.
>
>
> > How is the quality of annotations on exocentric videos using an HOI detector and how do the authors handle erroneous data?
>
> Thank you for your question! While it is inevitable to introduce some errors when using external tools, we implemented several filtering strategies to minimize these errors. For instance:
> - We filter generated narrations with low model confidence (Lines 242–246), which typically correspond to out-of-domain/egocentric-irrelevant videos.
> - We average HOI scores across frames (Lines 162–169) to alleviate the errors introduced by the HOI detector.
> - On the text side, we use techniques like text classification models to filter out non-visually alignable sentences (Lines 221–229).
>
> That being said, we acknowledge that the constructed dataset is not perfect and there is room for improvement. However, the experimental results demonstrate that the current techniques already achieved significant improvements over the baselines.

---

> > ### Comment · Reviewer_baF8 · 2024-11-25
> > **Reply to Author**
> >
> > The authors have adequately addressed my concerns, and my final rating is still borderline accept. This paper presents comprehensive experiments, demonstrating strong performance in multiple tasks, and has thoroughly optimized the entire pipeline to produce high-quality data. The reason I don't give a higher score is the lack of novelty. There have already been works on refining captions using LLMs and leveraging exocentric data for egocentric video understanding, and this work does not make a groundbreaking contribution.

---

> > > ### Author Response · Authors · 2024-12-02
> > > **Thank you!**
> > >
> > > We thank the reviewer for acknowledging our experiments, strong performance, and pipeline optimization. We will make our contributions clearer in the revised version, emphasizing how we identify and address the challenges of leveraging exocentric data for egocentric tasks with practical solutions that achieve notable improvements across action recognition, QA, and embodied tasks.

---

### Official Review · Reviewer_oLTq · 2024-11-03

**Soundness:** 3
**Presentation:** 3
**Contribution:** 3
**Rating:** 6
**Confidence:** 4

**Summary:**

Large-scale web-based datasets are commonly available; however, high-quality, ego-centric video data remains challenging to obtain. While datasets like EGO4D have been introduced, their scale still falls short in meeting the demands of scaling laws in Multi-modal Large Language Models (MLLMs).

In this work, they propose a method to extract ego-centric clips from web videos, primarily leveraging the HowTo100M dataset, which contains a substantial collection of instructional videos. This approach serves as an efficient way to retrieve valuable ego-centric clips from existing large-scale video data, expanding the resource pool for ego-centric research. Beyond common tasks explored in related ego-centric works, the author also conduct a series of experiments on HOI set and show inspiring improvement.

**Strengths:**

1. The motivation is clear: there remains a challenge in accessing high-quality ego-centric video data.

2. LaViLa-B demonstrates a clear improvement over both EgoVLP and EgoVLP V2 on the EK-100 MIR and EgoMCQ tasks.

3. The approach is straightforward and well-presented, making it easy to understand and practical for real-world applications.

4. Unlike previous ego-centric models, this paper proposes evaluating the model in a human-object interaction (HOI) setting.

**Weaknesses:**

1. The primary concern is the comparison and discussion of the advantages over retrieval-augmented methods, such as the Neural Data Server series. The critical aspect of EMBED lies in its need to access original data, denoted as X^{exo} in this work.

2. EMBED requires multiple offline models and must iterate through all candidates in a large corpus, such as HT100M in this work, making it time-intensive. Additionally, LANGUAGE NARRATION GENERATION relies on off-the-shelf LLMs, like LLAMA2. These steps involve substantial engineering efforts in data filtering, rather than being directly learned or optimized as part of the core model training.

3. From Tab4 we observe original noisy HT100M also boosts the performance a lot. This shows the improvement bring by EMBED is limited on UCF.

**Questions:**

1. The HT100M consists a large part of instruction video during data collection. So it is natural that this work can select suitable clip. If possible this work also works in other datasets like K600, Moments In Time etc.?

2. The HTM-AA dataset  is 247K videos, how about the samples number of HTM-EMBED in Tab.4 ?

3. Even though the EMBED data selection need to run only once for multiple experiments. Could you add the time comparison for LaViLa-B with LaViLa-B+EMBED.

---

> ### Author Response · Authors · 2024-11-23
> **Response**
>
> > The primary concern is the comparison and discussion of the advantages over retrieval-augmented methods.
>
> We agree that retrieving relevant exocentric videos based on their similarities to egocentric ones presents an alternative approach to data construction. However, we would like to highlight the following points:
>
> 1. EMBED and retrieval-based methods rely on different criteria for constructing data. Exploring Neural Data Server-style methods for egocentric learning could serve as a complementary direction to our work.
>
> 2. Compared with our method of computing HOI scores for clip selection, calculating similarity scores between videos for retrieval can be more complex and time-intensive.
>
> 3. We compared our approach with a RAG-based method (Xu et al., 2024) as shown in Table 1, demonstrating improved performance. To ensure a fair comparison, we also implemented an approach inspired by Xu et al. within our framework. Specifically, we retrieved relevant egocentric narrations based on exocentric ones and used Llama-3 to perform style transformations on the exocentric narrations using the retrieved narrations. However, this approach did not yield improvements, likely because retrieving relevant narrations that effectively support style transfer is challenging.
>
>
> | Method                   | EK100 MIR mAP  | EK100 MIR mDCG  | EgoMCQ Intra | EgoMCQ Inter |
> |--------------------------|-------|-------|-------|-------|
> | LaViLa-B + Xu et al.     | 34.4  | 33.6  | 59.5  | 94.0  |
> | LaViLa-B + Our Rephraser | 34.9  | 33.9  | 60.5  | 94.2  |
> | LaViLa-B + EMBED         | 36.0  | 34.9  | 61.3  | 94.5  |
>
>
>
> > EMBED requires multiple offline models and must iterate through all candidates in a large corpus, such as HT100M in this work, making it time-intensive.
>
> We acknowledge that EMBED requires several models for dataset construction. However, we would like to highlight that: 1) Our baseline models, such as LaViLa and Helping Hands, similarly rely on offline narration generation or HOI models for dataset processing. 2) The dataset construction occurs only during preprocessing and does not require further processing during training, ensuring that the training process remains relatively efficient.
>
>
>
> > Original noisy HT100M also boosts the performance a lot in Table 4 for exocentric tasks
>
> Our primary motivation is to construct egocentric-relevant data from exocentric sources. Thus, our methods are not specifically designed for cleaning HT100M but for adapting it to egocentric tasks. Consequently, it is expected that the improvements are primarily observed in egocentric tasks. However, the fact that our methods also achieve comparable or improved performance on exocentric tasks highlights their broader applicability.
>
>
>
>
> > If possible this work also works in other datasets like K600, Moments In Time etc.?
>
> Thank you for your suggestions! In Table 5, we actually demonstrated that applying our methods to other datasets (Kinetics-700, Something-Something v2, and COIN) can also achieve improvements over baselines.
>
> > Statistics of HTM-EMBED
>
> We list the statistics of our dataset in Line 255-260. Overall, there are around 202K videos and 2.4M video-language pairs.
>
> > Could you add the time comparison for LaViLa-B with LaViLa-B+EMBED?
>
> Because both LaViLa-B (Ego4D+HTM-AA) and LaViLa-B+EMBED (Ego4D+HTM-AA) are trained with ~2M exocentric video-language pairs, either from the original HTM-AA or our constructed one. Their training time is roughly equal and is ~1.5 days with 32 V100s for a base model.

---

### Official Review · Reviewer_rhcR · 2024-11-04

**Soundness:** 2
**Presentation:** 3
**Contribution:** 2
**Rating:** 5
**Confidence:** 4

**Summary:**

This paper presents EMBED (Egocentric Models Built with Exocentric Data), a novel approach for adapting exocentric video-language data to improve egocentric video representation learning. Although exocentric data is rich and varied, it lacks the close-up hand-object interactions and narratives central to egocentric perspectives.  EMBED bridges this gap by identifying specific video clips that emphasize hand-object interactions and pairing them with action-focused language narrations. Experiments show EMBED's effectiveness, achieving state-of-the-art performance with a 4.7% improvement on Epic-Kitchens-100 multi-instance retrieval and a 6.2% increase on EGTEA classification in zero-shot settings. Furthermore, EMBED enhances egocentric model performance on exocentric tasks and demonstrates strong generalization across diverse exocentric datasets, showcasing its potential to unify egocentric and exocentric video learning and capitalize on the unique strengths of exocentric data for egocentric applications.

**Strengths:**

(1) This paper iseasy to follow and  well-written.
(2) The proposed approach of using HOI to filter exocentric videos for egocentric learning is compelling, meanwhile the exo-to-ego rephraser and ego narrator prove effective. The paper provides extensive experiments across diverse datasets and benchmarks, offering strong support for its methodology.

**Weaknesses:**

The technical contribution of this work appears limited, as it primarily proposes a data filtering process for exocentric videos, utilizing pretrained LLMs to refine HowTo100 captions into an egocentric format and to generate additional egocentric narrations—an approach already widely explored in previous research. These methods seem more suited for the Ego4D workshop and competition, raising questions about their suitability for an ICLR submission.

Another key concern is the reliance on egocentric retrieval and classification benchmarks, which have been heavily used in recent years. It’s widely acknowledged that improvements on these benchmarks may not necessarily indicate a truly understanding of egocentric content or applicability in real-world scenarios. If the authors could provide additional experiments demonstrating consistent improvements on more challenging tasks, such as grounding or manipulation tasks, I would be inclined to reconsider my rating.

**Questions:**

Please refer to the weakness.

---

> ### Author Response · Authors · 2024-11-23
> **Response**
>
> > The technical contribution of this work appears limited, as it primarily proposes a data filtering process for exocentric videos.
>
> We would like to clarify the unique contributions of our work and our framework goes beyond a data filtering process. Specifically, as in our general response, we position our approach as an active video mining and tailored language generation process—a novel direction not previously explored.
>
> Furthermore, we compare our method with several existing models, demonstrating its effectiveness and highlighting its value to the research community.
>
>
> We will make these points more explicit in the revised version and are happy to incorporate comparisons with any additional references if provided.
>
>
>
> > Another key concern is the reliance on egocentric retrieval and classification benchmarks, which have been heavily used in recent years. It’s widely acknowledged that improvements on these benchmarks may not necessarily indicate a truly understanding of egocentric content or applicability in real-world scenarios.
>
> Thank you for raising this point!
>
> We would like to mention that egocentric retrieval and classification tasks are themselves representative of real-world applications. Additionally, most existing works on egocentric video learning (e.g., EgoVLP, EgoVLPv2, LaViLa) evaluate models on these benchmarks, and we follow their evaluation settings while also extending the evaluation to exocentric tasks for a more comprehensive analysis. Therefore, the existing evaluation results reflect the real-world applicability of our approach for these areas.
>
> That said, we agree on the importance of evaluating our model's applicability in broader settings. Following your suggestions, we evaluated our baseline model (LaViLa-B, Ego4D+HTM-AA) and our proposed model on egocentric QA tasks (EgoTaskQA [1] and ACQUIRED [2]) as well as an embodied task (Door-Open Human with RoboCLIP [3]) For egocentric QA tasks, we integrated our video encoder with a BART decoder and fine-tuned the model using LoRA. For the embodied task, we conducted a preliminary study by replicating the RoboCLIP setting, substituting its video encoder with our baseline or EMBED. As shown in the table below, our model consistently outperforms the baseline in these settings, demonstrating its generalizability.
>
> |          | EgoTaskQA-direct   | EgoTaskQA-indirect   | ACQUIRED-Ego | Door-Open Human RoboCLIP |
> |----------|---------------------|----------------------|--------------|--------------------------|
> | Baseline | 41.16              | 33.72               | 70.26        | 235.7                   |
> | Ours     | 43.54              | 34.49               | 72.11        | 255.0                   |
>
>
> [1] Jia et al., EgoTaskQA: Understanding Human Tasks in Egocentric Videos, NeurIPS 2022.
>
> [2] Wu et al., ACQUIRED: A Dataset for Answering Counterfactual Questions In Real-Life Videos, EMNLP 2023.
>
> [3] Sontakke et al., RoboCLIP: Generating Rewards using a Single Demonstration, NeurIPS 2023.
>
> We hope these results resolve your concerns and we are happy to address additional questions that you may have.

---

> > ### Comment · Reviewer_rhcR · 2024-11-25
> >
> > We appreciate the authors' efforts in conducting extensive experiments on these challenging tasks. Taking this into account, I have slightly adjusted my score to reflect a borderline position, leaning towards rejection (but close to the borderline). While this paper provides a comprehensive analysis of the proposed methods with good writing, the primary concern preventing a higher score remains the lack of novelty, as mentioned by other reviewers as well. I believe it would be more appropriate to engage in further discussion with the other reviewers before finalizing my evaluation. For now, my final score remains reserved.

---

> ### Author Response · Authors · 2024-12-02
> **Thank you!**
>
> We sincerely thank the reviewer for recognizing the strengths of our paper and engaging in a constructive discussion.
>
> Your suggestion to evaluate our approach in a broader context motivated us to conduct additional experiments on challenging QA and embodied tasks. These results further validate the effectiveness and value of our framework, and we will include them in the revised version.
>
> We will also make our contributions clearer, including systematically identifying the challenges of leveraging exocentric data for egocentric tasks and proposing practical, effective solutions that achieve significant improvements over robust baselines across action recognition, QA, and embodied tasks.

---

### Author Response · Authors · 2024-11-23
**General Response**

We sincerely thank all the reviewers for their time and their constructive reviews and suggestions. We are encouraged that the reviewers find that:
- Our proposed framework is compelling (rhCR) and practical for real-world applications (oLTq), and is a significant advancement in first-person video understanding by leveraging third-person video data (baF8).
- Our methods are effective, achieving significant performance improvements over existing models (rhcR, oLTq, baF8, XASQ).
- We provide extensive experiments across diverse datasets and benchmarks, conducting thorough ablation studies and diverse evaluations that offer strong support for our methodology (rhcR, baF8, XASQ).
- Our paper is easy to follow and well-written (rhcR, oLTq).

We would like to clarify the unique contributions of our work and stress that our framework goes beyond a data filtering or transformation process. Specifically, our contributions include:
1. Egocentric-Relevant Video Mining and Tailored Language Generation
- We mine new video clips using our proposed video temporal selection and spatial zoom-in techniques, which go beyond rewriting or re-generating narrations for existing video clips. This approach differs from existing works such as HowToCaption.
- Our primary goal is to tailor video data from one viewpoint to another, which differs from existing works focused on generating narrations for data augmentation or data cleaning.
- Considering these points, we view our approach not as a data filtering process but as an active video mining and tailored language generation process that has not been explored before.
2. Bridging Ego and Exo Data with Language Narrations and HOI Information
- Our approach integrates language narrations into the framework, distinguishing it from previous works like Ego-Exo, which primarily focus on video classification data and are hard to apply in video-language pretraining.
- Additionally, we explored simple and effective ways to use HOI information to bridge ego and exo data, outperforming a similar objective to Ego-Exo, as shown in Appendix C. This reinforces our hypothesis that previous methods are hard to apply in our setting and that new methods are necessary for effective video-language pretraining.
3. Performance Improvements and Empirical Contributions
- By combining our proposed methods that function in both the vision and language modalities, our method has yielded performance improvements, demonstrating its value for the research community.
- Compared with existing models such as EgoVLP, LaViLa, and Helping Hands, our method significantly outperforms these models on most of our evaluation benchmarks.

---

> ### Author Response · Authors · 2024-12-04
> **Summary After Discussion**
>
> We sincerely thank the reviewers for their thoughtful feedback and constructive suggestions, which have significantly strengthened our work.
>
> During the discussion period, we addressed key concerns raised by the reviewers, including:
> - Clarified Novelty: Emphasized that our framework integrates active video data mining and tailored language generation processes, going beyond data filtering or annotation refinement.
> - Broadened Evaluation: Added new experiments on challenging QA and embodied tasks, demonstrating the robustness and broad applicability of our method.
> - Ensured Fair Comparisons: Highlighted strict control of experimental settings, ensuring identical training strategies and data sources with baselines to isolate the contributions of EMBED.
>
> Our work systematically identifies the challenges of using exocentric data for egocentric representation learning and proposes practical, effective solutions that achieve substantial improvements over baselines across tasks. The reviewers' comments will be fully incorporated into the revised version.

---

### Meta-Review · Area_Chair_3h3b · 2024-12-24

**Metareview:**

The submission introduces a method to adapt large-scale exocentric videos to train egocentric video understanding models. The main contributions include an algorithm to select and refine close-up views of hand-object interactions, and a language narration rephraser, both of which enable the automatic construction of egocentric-style video datasets from large instructional video datasets. The proposed framework achieves strong performance on egocentric video benchmarks as well as other standard activity recognition benchmarks. The submission received mixed ratings after rebuttal, including three borderline accepts (6) and one borderline reject (5). Below I summarize the main strengths and limitations of the submission, according to the reviews (after rebuttal discussion) and my own reading of the submission:

*Strengths:*
- The proposed method is clearly presented and well motivated
- Comprehensive empirical evaluations and promising results

*Weaknesses:*
- Limited technical novelty
- The use of additional HTM data already brings performance improvement (although EMBED further pushes the performance)

The AC acknowledges that the submission offers a valuable empirical study on bridging the exo- and ego-centric video domain gap via human-object interaction regions. However, the AC also shares similar concerns with reviewers rhcR, baF8, and XASQ on the "technical novelty" aspect of the submission. While such "technical novelty" is not always required for a high-quality ICLR paper, the submission's focus on bridging the exo- and ego-centric domain gap raises potential questions on (1) how generally applicable the proposed framework would be for applications of broader interest for the ICLR community; (2) if the insights obtained by cropping HOI regions, and other heuristics introduced by the submission, could be extended to other video/visual understanding problems.

After reading the submission, the reviews, and the discussions, the AC has reservations for the two questions above, and therefore cannot recommend the submission to be accepted by ICLR 2025.

**Additional Comments On Reviewer Discussion:**

The authors did a solid job addressing the following questions:
- Evaluations on even more benchmarks
- Comparisons with retrieval augmented methods
- Time efficiency with respect to baseline methods
- Model and data release
- Performance on SSv2

After the discussion phase, there were remaining concerns on the "technical novelty" (XASQ, baF8, rhcR). A minor concern was on the use of HTM, which helps, but does not solely contribute to, the performance of the proposed framework. The AC shares a similar concern on the "novelty" aspect of the submission, with rationale outlined above.

---

### Decision · Program_Chairs · 2025-01-22

Reject